# Early Subcellular Hepatocellular Alterations in Mice Post Hydrodynamic Transfection: An Explorative Study

**DOI:** 10.3390/cancers15020328

**Published:** 2023-01-04

**Authors:** Mohd Yasser, Silvia Ribback, Katja Evert, Kirsten Utpatel, Katharina Annweiler, Matthias Evert, Frank Dombrowski, Diego F. Calvisi

**Affiliations:** 1Institut fuer Pathologie, Universitaetsmedizin Greifswald, Friedrich-Loeffler-Str. 23e, 17475 Greifswald, Germany; 2Institut fuer Pathologie, Universitaetsklinikum Regensburg, 93053 Regensburg, Germany

**Keywords:** hydrodynamic transfection, hydrodynamic gene delivery, gene therapy, hepatocellular carcinoma, electron microscopy

## Abstract

**Simple Summary:**

Our electron microscopic (EM) findings indicate that hydrodynamic based plasmid transfer to the hepatocytes possibly occurs via active or passive endocytosis with the plasmid entrapment in membrane bound multiple vesicle and vacuoles. Our ultramorphological study on the vacuole containing mouse hepatocytes did not show disrupted plasma membrane or other signs of cellular damage, thus keeping out the mechanism like membrane poration, but cannot be entirely excluded at the micro level. Independent of the plasmid itself or its enclosed components, we have also found the vesicle formations is a non-specific process. Therefore, it remains to be clarified as how the active or passive mechanisms are involved, leading to the vesicle generation and DNA-entry to the nucleus of the hepatocytes.

**Abstract:**

Hydrodynamic transfection (HT) or hydrodynamic tail vein injection (HTVi) is among the leading technique that is used to deliver plasmid genes mainly into the liver of live mice or rats. The DNA constructs are composed of coupled plasmids, while one contains the gene of interest that stably integrate into the hepatocyte genome with help of the other consisting sleeping beauty transposase system. The rapid injection of a large volume of DNA-solution through the tail vein induces an acute cardiac congestion that refluxed into the liver, mainly in acinus zone 3, also found through our EM study. Although, HT mediated hydrodynamic force can permeabilizes the fenestrated sinusoidal endothelium of liver, but the mechanism of plasmid incorporation into the hepatocytes remains unclear. Therefore, in the present study, we have hydrodynamically injected 2 mL volume of empty plasmid (transposon vector) or saline solution (control) into the tail vein of anesthetized C57BL/6J/129Sv mice. Liver tissue was resected at different time points from two animal group conditions, i.e., one time point per animal (1, 5, 10–20, 60 min or 24 and 48 hrs after HT) or multiple time points per animal (0, 1, 2, 5, 10, 20 min) and quickly fixed with buffered 4% osmium tetroxide. The tissues fed with only saline solution was also resected and fixed in the similar way. EM evaluation from the liver ultrathin sections reveals that swiftly after 1 min, the hepatocytes near to the central venule in the acinus zone 3 shows cytoplasmic membrane-bound vesicles. Such vesicles increased in both numbers and size to vacuoles and precisely often found in the proximity to the nucleus. Further, EM affirm these vacuoles are also optically empty and do not contain any electron dense material. Although, some of the other hepatocytes reveals sign of cell damage including swollen mitochondria, dilated endoplasmic reticulum, Golgi apparatus and disrupted plasma membrane, but most of the hepatocytes appeared normal. The ultrastructural findings in the mice injected with empty vector or saline injected control mice were similar. Therefore, we have interpreted the vacuole formation as nonspecific endocytosis without specific interactions at the plasma membrane.

## 1. Introduction

HT employs hydrodynamic pressure to facilitate intracellular gene transfer [1,2,3,4], is one of the latest murine non-viral gene transfer model with high efficiency in liver cancer research. It is a more favorable transfection method for liver-specific studies as 90% of transgene products are predominantly found in the liver, albeit lungs, kidney, spleen and heart detect lower expression levels [2,5]. Because of its compelling transgene expression in the liver, it has been successful as a frontline approach in identifying liver-specific oncogenes and tumor suppressor genes that contribute to liver tumor initiation and progression [6,7].

In the beginning, HT was used only to deliver genes in mice [2]; however, more recently, it has broadened its dimension from delivering far-ranging substances like DNA [2,8,9], RNA [10,11,12,13,14,15,16], artificial chromosomes [17], proteins [8], small interfering RNA (siRNAs) [18], polymers [19,20], and other compounds [21]. The elegance of HT relies on the fact that various substances can be delivered without receptor-ligand interaction involvement, and it is regularly maturing, since its unravelling in both of its approaches to transport cell membrane impermeable molecules, a fundamentally most useful method for intracellular delivery in rodents [22] and its appositeness in larger animals like pigs [23] and dogs [24,25].

Competent, in-vivo transgene expression is paramount for studying functional disease treatment approaches in the liver as its hub for numerous metabolic pathways and thus involved in associated diseases. Although the HGD (hydrodynamic gene delivery) mechanism in hepatocytes has been well studied [1,26,27], the uptake mechanism of the plasmid into hepatocytes after HT remains unclear. One can imagine that the hepatocytes are mainly targeted because of their close association with the sinus endothelium. Indeed, it has been shown that HT induces a hydrodynamic force leading to the permeabilization of the fenestrated sinusoidal endothelium [1]. This allows nucleic acids entry to the hepatocytes, the movement to the nucleus followed by the closure of membrane transient pores in a short period [26,27,28], and the facilitated expression. Therefore, here we want to provide an updated overview of the hydrodynamic based transfection and discuss our preliminary findings of mechanistic transfection possibilities in the mouse liver.

### 1.1. Principle for HT in Mice

In Principal, a rapid injection in less than 10 s [29] of a larger volume, proportionate to 10% body weight containing plasmid DNA has been injected into the mouse tail vein [6,7,30]. First reported by Budker et al. [31], this hydrodynamic pressure expands the liver and enforces hepatocytes membrane invaginations to grant the macromolecule entry [28]. Specifically, such tension heightens the hepatocytes permeability; the literature suggests, it transiently generates numerous membrane pores [26]. As a result, about 40% of hepatocytes in zone 3 of the liver acinus can be transfected with the gene of interest [26]. In rats, Zhou et al. has previously showed, that the highest level of a reporter gene (Lac Z; pCMV-Lac Z containing β-galactosidase gene) are expressed in the liver [32] compared to other organs like: lung, heart, kidney and spleen (after determining histochemicaly the β-galactosidase gene expression in different organs, post hydrodynamic gene delivery). Such observations are very similar to those in mice, as reported by Liu et al. [2] and Zhang et al. [8] found that, among the organs expressing the transgene, liver showed the highest level of gene expression.

Also, summarized in the Figure 1a, the rapid injection of a large volume, may leads to a rapid congestion in the mice heart, that reflux mainly into the liver (thick lines) and can eventually permeabilize the plasma membrane of the hepatocytes, with ultimate reaching near to the liver—acinus zone 3, due to the dynamic force carried by the pressurized solution.

Notably, without adopting a viral vector, such transfection efficiency is apical in an in-vivo setting with long-lasting expression (4 weeks) [33,34,35,36,37,38,39], reaching therapeutic levels [40,41,42], and has been used in augmenting lenti- [43], adeno- [44,45], and adeno-associated [46] viral vectors transduction efficiency.

Studies have performed earlier, shows the efficient, long-term gene expression to the mouse liver cells using a hydrodynamics-based procedure. For example: Zhang et al. [33] systematically characterize that serum level of human α1-antitrypsin (hAAT) gene as a reporter can reach as high as 0.5 mg/mL by a simple tail vein injection of 10–50 μg plasmid DNA into mouse, with a detectable level (2–5 μg/mL) persists for at least 6 months (with hAAT gene in mouse liver remained active, 6 months after transfection). Liu et.al. [2] by using luciferase cDNA and β-galactosidase reporter gene have demonstrated that, post hydrodynamic transfection, consisting single tail vein injection of 5 μg of plasmid; liver shows as high as 45 μg of luciferase protein per gram of its weight. These studies suggest, liver as an important target organ for hydrodynamic based gene transfer, may be due to its peculiar anatomical position and its large capacity and therefore, makes HT a valuable tool for therapeutic purposes in the whole animals.

Although injection leads to cardiac congestion, the liver injury heals within 1 week [26]. The long-term sustained expression of a transgene in the hepatocytes for example, can be achieved by using the Sleeping Beauty (SB)—transposon system, precisely by transposing the expression cassette from a plasmid into the genome by the aid of transposase enzyme [30,47,48,49]. Indeed, as a non-viral vector system, SB transposon system work in consensus with the hydrodynamic delivery procedure as stably integrated system and thus determine the extent of gene delivery to the mouse liver [50,51].

### 1.2. Liver Targeted Hydrodynamic Gene Therapy: Recent Advances & Experimental Applicability

Several features are unique to the liver, which contribute to the high transgene expression in this organ, including the presence of fenestrated sinusoids, absence of a basement membrane, large surface area of hepatocytes facing the lumen, proximity to the IVC, low blood pressure, high capacity of hepatocytes for gene expression and large surface area of hepatocytes facing the lumen. Because of its simple nature and minimal technical requirements, it has become the first choice for studying relevant interests to various sys-tematic liver diseases that can alter the liver phenotype by altering certain gene expressions. It has been used to express haemophilia factor VIII [52], alpha-1 anti-trypsin [33], cytokines [53], hepatic growth factors [54], and erythropoietin [2] in mouse and rat models. The gene expression level with the best combination of expression vectors and regulatory elements has reached or exceeded the physiological level [3]. It is important to understand that together with the efficient genetic delivery system, its long-term expression is also critical in achieving the therapeutic effects. In this regard, the combination of HT and transposable elements can be considered as natural gene delivery method for stable genomic insertion, long-term and efficient transgene expression and inexpensive to manufacture—an important amalgamation regarding the implementation of future clinical trials [55].

HT also stands outstandingly in the treatment of liver fibrosis [56,57]. HGD of matrix metalloproteinase-13 gene reduced liver fibrosis and showed mentionable gene therapy efficiency [58]. It has been known that Hepatocellular carcinoma (HCC) has a high mortality rate. To study more precisely and explore the underlying HCC mechanisms, researchers developed various genetically engineered mouse (GEM) models [59]. However, developing GEM models is time-consuming and expensive; to fill such caveats, HT-mediated gene transfer methodology has been successfully applied in generating various HCC mouse models coupled with the sleeping beauty transposase system [7]. Further, it has been reported to be used in the establishment of small animal disease models for hepatitis, liver fibrosis, liver cancer, sepsis, and lipidemia [11,60,61,62,63,64,65] to the development of multiple tumor mouse models in the liver, kidney, and lung [66].

Thus, the emerging perspectives of HT in liver targeted gene delivery can be clearly understood, and with the advent of gene editing methods, its applicability in the in-vivo research field is currently in focus.

### 1.3. Gene Knock-Down in Liver

Functional gene approaches are indispensable in studying liver diseases and thus constitutes a basic mode. For example, on one hand sustained transgene strategy has been reported, where liver-specific high levels expression can be achieved, without using the integration systems [67], whereas the specific gene knockdown by RNA interference can be achieved on the other hand using HT [21] that makes it a powerful tool for liver-specific gene knockdown studies. It is also important to understand that, due to the lack of efficient delivery methods for small interfering RNA (siRNA) or short hairpin RNA (shRNA) vectors, RNA interference is limited in vivo [68].

Researchers, with the help of HT approach in vivo, silenced Fas [5,69], mdr1a/1b gene expression [70]. To make the HT-mediated strategy more optimal, siRNA-expressing adenoviral vectors were used to improve the knockdown effects of gene expression in the murine liver [71] and demonstrate long-term RNA interference in vivo [72,73].

### 1.4. EM and Hydrodynamic Transfection

Electron microscopy (EM) capability of greater magnification and resolving power to visualize more refined details of sub-cellular components [74] have been advantageous to get insight into the distribution of cell membrane pores and transported vesicles distribution that composes the fundamental chemistry for hydrodynamics mediated transfection. Little is known about the structural features and distribution of membrane pores at the sub-nanometer scale [75]. In a previous study, shortly after HT, EM shows the existence of transient membrane pores in hepatocytes that could be the plasmid DNA entry [26] and allows direct transfer of substances into cytoplasm without endocytosis [76]. In this regard, we are also discussing our results here.

## 2. Materials and Methods

### 2.1. Animal Studies

Housing of the animals was in accordance with the guidelines of the Society for Laboratory Animal Service and the German Animal Protection Law. Highly inbred 6 weeks old male C57BL/6J/129Sv mice (n = 20; 20–25 g body weight) were purchased from Charles River Laboratories (Sulzfeld, Germany). The housing of the animals was in accordance with the guidelines of the Society for Laboratory Animal Service and the German Animal Protection Law.

### 2.2. Hydrodynamic Tail Vein Injection and Tissue Processing

The constructs in these experiments included 10 µg of empty pT3-EF1α-vector (without a gene sequence), which were diluted in a ratio of 1:25 in 2 mL of 0.9% NaCl. The control mice only received saline solution. Plasmids were rapidly (less than 10 s) injected into the mouse lateral tail vein. In the first series of experiments (animals no. 1–11), we determined only one-time point for one animal (after 1, 5, 10–20, 60 min or 1 and 2 days respectively as stipulated in the Table 1 and also showed pictorially in the Figure 1b). These animals were sacrificed via cervical dislocation, and the liver tissues were immediately taken for fixation. While, in the second series (animals no. 12–20), we determined two, three or four time points per animal. These mice had to be anaesthesized (50–100 mg/kg body weight ketamine, 10 mg/kg body weight xylazine) before tail vein injection.

After median laparotomy and liver exposure, the respective liver lobes were clamped, and tissue pieces were immediately taken for the fixation. Samples of the respective liver tissue were cut with a razor blade to specimens of 2 mm^3^ pieces, immediately fixed in 4% osmium tetroxide (buffered in 0.2 mol cacodylate, pH 7.4) for 2 h, and buffered in cacodylate afterward.

### 2.3. Ultrastructural Analysis

The osmium-fixed liver tissue pieces were embedded in Glycidether 100 (Serva, Heidelberg, Germany), cut with diamond knives (Science Service GmbH, Munich, Germany) with a Leica ultratome (Leica EM UC7, Leica Biosystems, Wetzlar, Germany) to 500 and 750 nm thick semi-thin slides. While, the 70–90 nm ultrathin sections were stained with two-step staining procedure consist of uranyl acetate (Merck, Darmstadt, Germany) and lead citrate (Sigma-Aldrich, Darmstadt, Germany) as the standard routine contrasting technique for electron microscopy and examined with Libra 120 electron microscope (Carl Zeiss, Jena, Germany). The remaining liver tissue was fixed in formalin and embedded in paraffin. Paraffin slides of 1–2 µm thickness were serially cut and stained with H&E and the periodic acid Schiff (PAS) reaction.

## 3. Results

In comparison to unaltered hepatocytes before injection (Figure 2), after 1 min, hepatocytes near the central venule in acinus zone 3 reveal some small membrane-bound vesicles in the cytoplasm and perisinusoidal space of Disse are filled with erythrocytes and blood particles (Figure 3). After an interval of 10 to 20 min, the vesicles increased in size to form vacuoles and also increased in numbers (Figure 4). These vacuoles often found in closeness with nucleus, optically empty without containing any electron-dense material in them. Some of the other hepatocytes reveal signs of damage at cellular level which comprises of swollen mitochondria, dilated Golgi and endoplasmic reticulum with disrupted plasma membrane. Of note, most of the hepatocytes are appeared normal and all findings were similar post injecting the empty vector or the saline solution.

After 60 min, almost all vesicles and hepatocellular alterations vanished (Figure 5). Indistinguishable trends were observed for the later time-points (post 60 min).

Zone 3 is located furthest from the portal triad and closest to the central vein, thus due to its distinctive location, is more susceptible to ischaemic damage [77]. Therefore, although zone 3 has the lowest perfusion due to its distance from the portal triad, but such farther travel is possible due to the forceful cardiac congestion induced by hydrodynamic transfection and our data also suggest that the reflux of large volume of transfected solution as a result of possible acute cardiac congestion goes mainly to acinus zone 3 of the liver.

These prolific observations disclose by the assistance of electron microscopy through our study will be useful in further advancing our understanding towards important metabolic role played by Liver, as zone 3 plays the largest role in detoxification, biotransformation of drugs, ketogenesis, glycolysis, lipogenesis, glycogen synthesis and glutamine formation [78].

Further informative evidences indicated from our present study, that the vesicles containing hepatocytes mostly shows intact cell borders and therefore, pave the way to further conceptualize the idea of hydrodynamic transfection mediated membrane transport.

## 4. Discussion

### 4.1. Hydrodynamic Transfection and Membrane Transport

Bilayer phospholipid molecules interspersed with protein molecules provide semi or selective permeability. The outer surface comprises tightly packed hydrophilic (or water-loving) polar heads and middle hydrophobic (or water-fearing) nonpolar tails consisting of fatty acid chains. In this way, a 6 to 10 nanometers thick cell membrane acts as a semipermeable barrier and maintains homeostasis by providing selective to and from traffic in both directions.

Of note, the plasma membrane is the barrier for nucleic acid delivery to hepatocytes. On the one hand, applied hydrodynamic pressure still physically aids in overcoming this barrier. However, this lone effort seems still insufficient to gain DNA entry and subordinate over the innate molecular nature of the membrane through either active or passive transport [79,80]. Further, it has been documented that HGD enlarges membrane pores due to the high pressure it generates (up to approximately 100 nm in diameter) [81]. Such setup facilitates the passage of exogenous DNA across the liver capillary endothelial cells to be taken up by the hepatocytes [26]. After DNA delivery, the pores are resealed within a few minutes [1], which hints toward a coordinated effort in delivering the DNA molecule across the membrane.

However, the mechanisms responsible for plasmid incorporation into hepatocytes remain unclear and vesicle formation already shown by Crespo et al. [28] and enhancement of membrane pores reported by Zhang et al. [26] pose the possible mechanism of entry via hydroporation. Here, we mainly review the principle, developments, and current applications in liver disease studies and discuss the transfection methodology from a transmission electron microscopy perspective in small animals and its future challenges and clinical potential.

### 4.2. Passive or Active Membrane Transport

The plasma membrane remains in continuous motion and a site of constant recycling where small, moderately polar molecules can passively diffuse across the cell membrane without requiring cellular energy expenditure [82,83,84]. The unassisted diffusion of very small or lipid-soluble particles is called simple diffusion, while the assisted process is known as facilitated diffusion. In this effort, transmembrane channel proteins create diffusion friendly openings for molecules to move through.

However, the number of natural molecules known to diffuse across the cell membrane passively is surprisingly limited [85], and the cell relies mainly on the membrane transporters, but on the expense of energy in the form of ATP to deliver compounds that are more polar in nature such as amino acids, peptides and nucleosides. It is interesting to note that approximately 10% pool of human genes take care by the transporters, thus accentuating its functional significance [86].

This broadly constitutes the “endosomal release”, where the cargo is transported across the membrane to access the cytoplasm.

### 4.3. Endocytosis

Endocytosis is the active transport process involving vesicle formation from the plasma membrane and constitutes one way by which materials can enter the cell from the extracellular environment. It is essential for eukaryotic cells and well characterized in mammals [87] to take various forms depending on the cellular machinery involved, ranging from specific receptor-mediated endocytosis to less selective pinocytosis.

Most importantly, HT is a very rapid and straight forward technique and can be possibly engrossing with respect to our present findings. The membrane pores can vary in their diameters; thus, the facilitative mechanism lets the traversing molecules pass through [85,88]. For example, the membrane pores might facilitate the intracellular delivery of small molecules (<4 kDa), while the endocytosis might induce the uptake of macromolecules (>4 kDa) [89,90].

Among receptor-mediated, the clathrin-mediated endocytosis forms, the major internalization route into a eukaryotic cell [91], propelled by the mechanical force needed to detach a vesicle (30–100 nm) from the plasma membrane on the cytoplasmic side and also involves the participation of dynamin which is fundamental for vesicle scission [92,93]. Indeed, electron microscopy first observed clathrin-coated vesicles in the early 1960s [94,95,96] and is the best-understood mechanism. On the other hand, the non-clathrin mediated endocytosis constitutes many forms driven by endophilin or caveolin endocytic proteins [97]. These cholesterol binding caveolins provide aid on the plasma membrane to the cholesterol and sphingolipid-rich nano-domains assembled by caveolae [98]. In more simple way, the caveolins are cholesterol-binding integral membrane proteins assembled in caveolae, where they colocalized with many receptors [99]. Other endocytosis forms are: Pinocytosis (small endocytic vesicles, receptor-independent), Transcytosis (receptor-mediated, coupled with exocytosis to transport cargo across the interior of a cell), and Macropinocytosis (a complex pinocytosis form) that facilitate cell to imbibe nearby cargo also consisting of extracellular fluid into the large endocytic vesicles recognized as macropinosomes [100,101,102].

However, disregarding to initial uptake process, the endocytosed molecule is subject to endosomal trafficking in transport vesicles through the intracellular space. Although significant propositions have been made in the predictive designs of intracellular delivery technologies, the repertoire is steadily expanding. In this quest, our results also suggest the possible role of assisted endocytic transport across the hepatocellular membrane, which can also be further understood in Figure 6 summarized here. Briefly, the classical endocytosis could play a major role in the vesicle trafficking across hepatocellular membrane (Figure 6a,b) with additional possibility of the membrane poration (Figure 6c). Particularly, these temporary membrane porations are results of a multistep process comprises of (i) Compress phase: where the quick compression generated by a sharp rise in the venous pressure can interchangeably compress the extracellular fluid with (ii) Expansion phase: leads to its expansion during the short period of temporary liver congestion and ultimately (iii) Transient pore: formation across the hepatocellular membrane. Zhao et al. has also reasoned such idea as a result of disruption at the cell membrane in his review [103].

### 4.4. Advantages and Challenges of HT for Preclinical Application and Gene-Drug Discovery

Mechanistically, the most favorable advantage of hydrodynamic based transfection is that the delivered molecules do not need packaging. In fact, this method can also successfully transfect bacterial artificial chromosomes (BAC) as large as 150 kb [17,104]. It is one of the most commending method that makes liver the prime site as an exploratory organ for clinical and therapeutic purposes, because it can bypass the targeted step costs when live animals are used [3]. From the gene-drug discovery perspective, the advantage of HT in studying a gene’s therapeutic function is tremendous. Different constellations of the DNA fragments, such as multiple specific regulatory expression sequences, can be efficiently transfected hydrodynamically into animals carrying a target disease. Furthermore, the repeated gene transfers and different regimens can be fine-tuned to maximize the therapeutic outcome. The transgene introduction in mice shows that HT is safe and effective [95,96] and that developing a safer and more efficient in-vivo gene delivery is imperative. Being a non-viral delivery method, HT has vast potential in this regard but still has considerable challenges like lower gene-delivery efficiency that requires for the therapeutic levels [4,56], broader applicability in larger animals that will help in paving the way for accelerated development of this simple and cost-effective technique in humans [28,56,105,106]. Interestingly, Hackett et al. hypothesized that ‘impulse’ rather than ‘pressure’ could be a critical determinant of the effectiveness of hydrodynamic delivery in larger animals and thus propose some variance in delivery strategies [29]. To a large extent transgene incorporation by HGD is safe and effective in mice with commensurate success in rats, but its application in larger animals remains a challenge, nevertheless the field is continuously evolving with newer modifications [104].

For now, it has been widely reported that HT does not have a significant concussion on the mouse in general; it exerts only temporary cardiac dysfunction, a sharp rise in venous pressure [1], and temporary liver congestion with no significant destruction of liver cells [105] and elevation of aminotransferase (ALT) that recovers within 2–3 days with no signs of hepatic failure [26,106]. Further, the image-guided procedure in HT-based gene delivery is one of the recent developments in the field that aims to develop for reducing the injection volume and directly supports its futuristic clinical applicability [18,56,107,108,109]. With these constraints, HT is a unique tool with its wings spread for use in any mouse strain to establish in different animal models.

## 5. Conclusions

Our findings based on the ultra-morphology largely stipulate that the hydrodynamic based plasmid transfer to the hepatocytes may artfully carried out via passive or active endocytosis with entangling of the plasmid, amassed into membrane bound vesicles and vacuoles. Since, in our findings, we have not seen any clues of the damage at cellular level in the vacuoles containing hepatocytes, therefore we are ruling out the possibility of mechanisms like membrane poration, but are not entirely excluding it. Also, in our findings the formation of vesicles is found to be independent of the plasmid, so the question is still remaining to be further clarified, related to defining the precise mechanism involving the vesicle generation and method of DNA entry to the nucleus of hepatocytes. Of note, the mammalian cell plasma membrane is a baroque of collective lipid with protein species continually undergoing endocytosis and exocytosis. The main confrontation for in-vivo gene delivery is to enhance efficiency with minimal tissue damage and on the other hand, DNA complexes are mild complexes that can successfully integrate through the membrane mechanistically. But, it is fascinating and still evolving to understand the precise mechanism behind the in-vivo delivery of the macromolecules like DNA, RNA, proteins or other membrane impermeable compounds. However, regarding the probable mechanism for the hepatocyte specific delivery, we collectively believe that the endocytosis-mediated vesicle transportation has the upper edge.

When the hydrodynamics-based transfection is engaged in a whole animal, the physiological proceedings have been reiterated in drawing sincere conclusions. In this regard, understanding the liver biology for its cargo delivery approach into hepatocytes could offer its potentiality for other organs as well, therefore continuing its advancement in both primary and clinical research. For the near future, further analysis of the evolving HT mechanism and its physiological implications for cargo-selective trafficking will certainly help us in better understanding its diverse mechanism. Furthermore, high-throughput discovery research and computational modelling could establish the regulatory association, linking endocytosis components with membrane physiology.

## Figures and Tables

**Figure 1 cancers-15-00328-f001:**
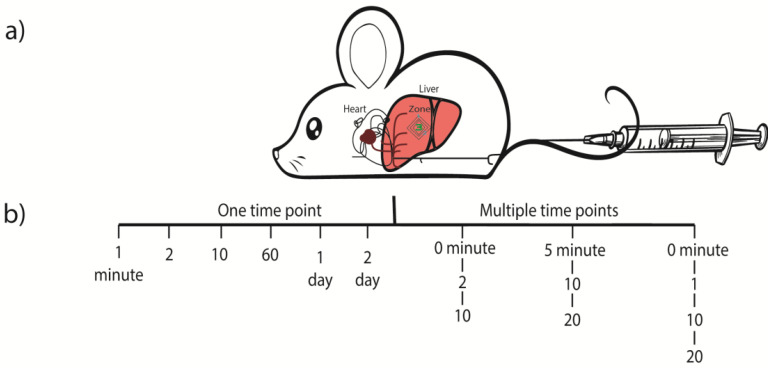
Schematic diagram showing (**a**) Hydrodynamic injection into the tail vein of C57BL/6J/129Sv mice. Briefly, 2 mL volume of empty vector or saline solution (control) was hy-drodynamically injected into the tail vein of anaesthesized C57BL/6J/129Sv mice. (**b**) Liver tissue was resected at different time points and quickly fixed with buffered 1% osmium tetroxide. For electron microscopic evaluation, ultrathin sections were cut from glycidether embedded blocks.

**Figure 2 cancers-15-00328-f002:**
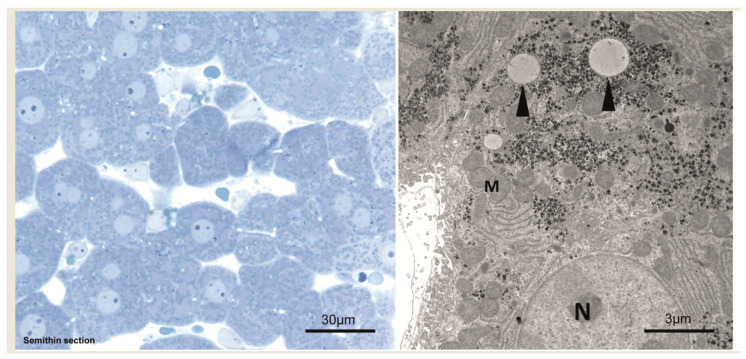
Unaltered hepatocytes before hydrodynamic injection. Left side: Semithin section, Right side:Ultra-structure of unaltered hepatocyte before hydrodynamic injection. N—nucleus; M—mitochondria; arrowheads—lipid vacuoles.

**Figure 3 cancers-15-00328-f003:**
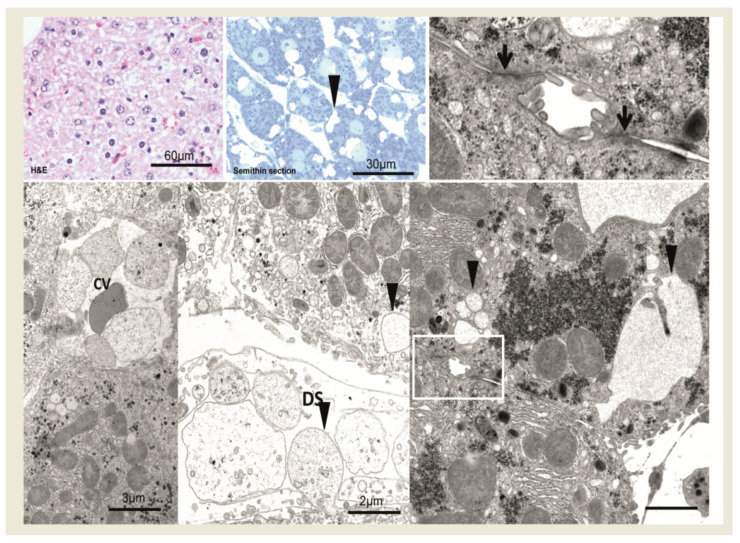
After 1 min, central venules (CV) and perisinusoidal space of disse (DS) are filled with erythrocytes and blood particles (H&E staining). In semithin section and at the ultrastructural level, hepatocytes near the central venule in acinus zone 3 reveal small and larger membrane-bound vesicles in the cytoplasm (arrowheads). Cell membranes and tight junctions (magnified box, lower right and arrows, upper right) are intact. Scale bars are indicated in the respective image.

**Figure 4 cancers-15-00328-f004:**
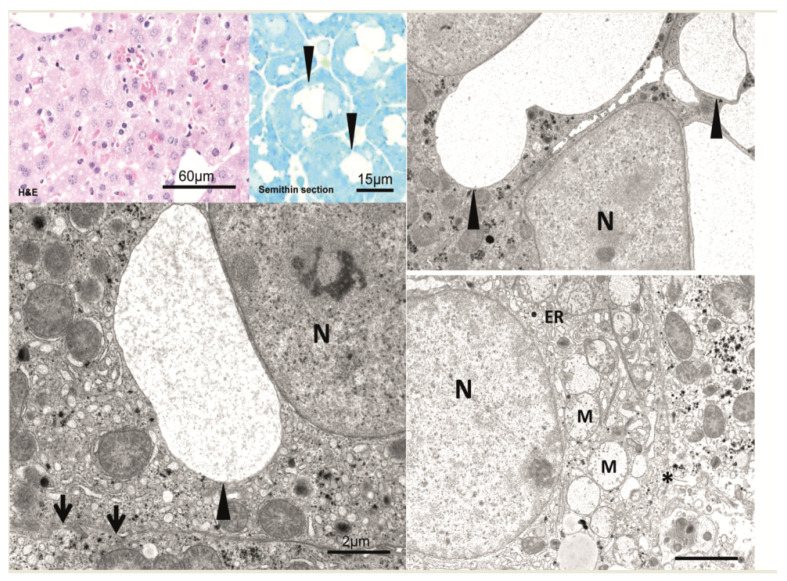
After 10 to 20 min, vesicles increase in size and number, often forming vacuoles (arrowheads). Vacuoles can usually be found in proximity to the nucleus (N). They are optically empty and contain no electron-dense material. Vesicle-containing hepatocytes show intact cell borders (arrow). In contrast, some but other hepatocytes reveal signs of cell damage, i.e., swollen mitochondria (M), dilated endoplasmic reticulum (ER), and disrupted plasma membranes (asterisk). All findings were similar after the injection of the empty vector and the saline solution as well. Scale bars are indicated in the respective image.

**Figure 5 cancers-15-00328-f005:**
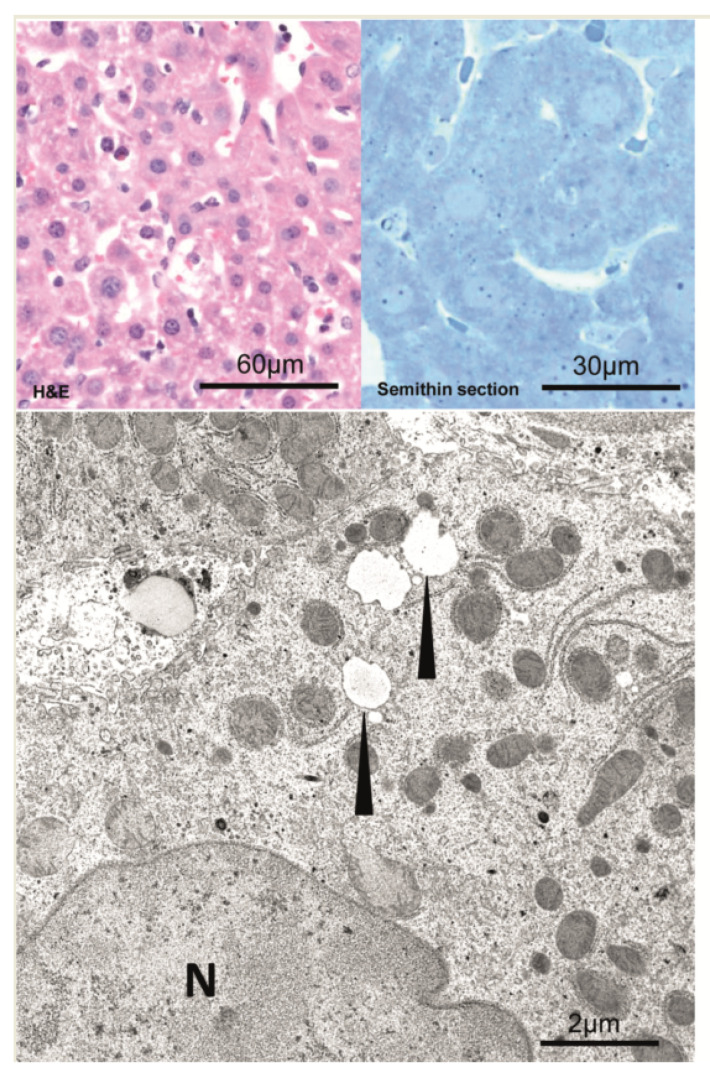
After 60 min, almost all vesicles (arrowheads and hepatocellular alterations vanished. N—nucleus. Scale bars are indicated.

**Figure 6 cancers-15-00328-f006:**
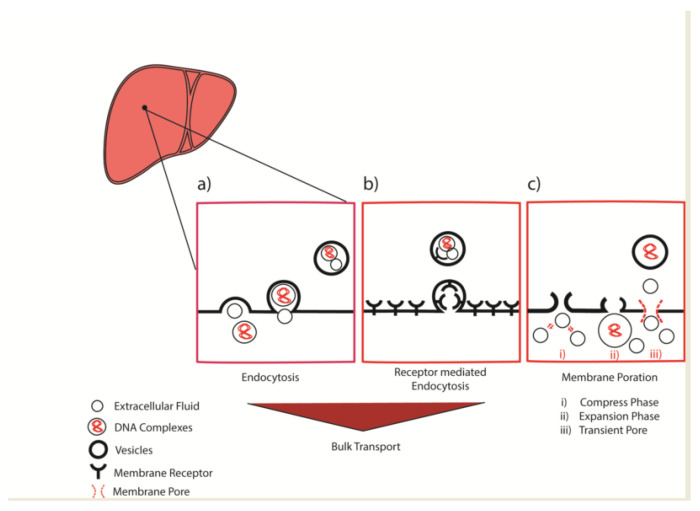
Schematic diagram representing (**a**) Endocytosis (**b**) Receptor mediated endocytosis as the bulk intercellular vesicle transport or (**c**) Membrane poration across the hepatocellular membrane.

**Table 1 cancers-15-00328-t001:** Hydrodynamic injection and liver tissue preparation time points.

Animal Numbers	Time after Hydrodynamic Injection to Liver Tissue Preparation and Immediate Fixation in Osmium Tetroxide							
	m: min d: day							
Empty plasmid								
One time point/animal								
1		1 m						
2		1 m						
3				5 m				
4					10 m			
5					10 m			
6							60 m	
7								1 d
8								1 d
9								2 d
10								2 d
11								2 d
Multiple time points/animal								
12	0 m		2 m		10 m			
13	0 m	1 m			10 m	20 m		
14				5 m	10 m	20 m		
15				5 m	10 m	20 m		
16				5 m	10 m			
17				5 m	10 m	20 m		
Control (0.9% NaCl)								
Multiple time points/animal								
18				5 m	10 m	20 m		
19				5 m	10 m			
20				5 m	10 m			

## Data Availability

Not applicable.

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
