# Peer review of "Early Subcellular Hepatocellular Alterations in Mice Post Hydrodynamic Transfection: An Explorative Study"

_cancers, 2023, doi:10.3390/cancers15020328_

Round 1

Reviewer 1 Report

Yasser et al. have crafted a very nice, easy to read and understand paper on hydrodynamic transfection for delivery of nucleic acids (and other macromolecules?) to mouse liver. The manuscript is described as updated overview of the Hydrodynamic based Transfection. Accordingly, I suggest a modest increase in citation and explanation. 

1) Introduction: The earliest papers in 199 were reference 12 and Zhang G, Budker V, Wolff JA. High levels of foreign gene expression in hepatocytes after tail vein injections of naked plasmid DNA. Hum. Gene Ther 1999;10:1735–1737. A very comprehensive review on step by step procedures with pitfalls, etc. is: Bell JB, Podetz-Pedersen KM, Aronovich EL, Belur LR, McIvor RS, Hackett PB. Preferential delivery of the Sleeping Beauty transposon system to livers of mice by hydrodynamic injection. Nat Protoc. 2007;2(12):3153-65. doi: 10.1038/nprot.2007.471. 

There were two publications back to back on dogs; if anything the Aronovich paper is more in line with this review (but both should be cited together) Aronovich EL, Hyland KA, Hall BC, Bell JB, Olson ER, Rusten MU, Hunter DW, Ellinwood NM, McIvor RS, Hackett PB. Prolonged Expression of Secreted Enzymes in Dogs After Liver-Directed Delivery of Sleeping Beauty Transposons: Implications for Non-Viral Gene Therapy of Systemic Disease. Hum Gene Ther. 2017 Jul;28(7):551-564. doi: 10.1089/hum.2017.004.

2) Section 1.1 (and 2.2): The duration of injection is critical; a good figure for this is in Hackett PB Jr, Aronovich EL, Hunter D, Urness M, Bell JB, Kass SJ, Cooper LJ, McIvor S. Efficacy and safety of Sleeping Beauty transposon-mediated gene transfer in preclinical animal studies. Curr Gene Ther. 2011 Oct;11(5):341-9. doi: 10.2174/156652311797415827. 

3) Section 1.1 An alternative to hydrodynamic pressure as the basis for HT is hydrodynamic impulse, which explains why HT has never really worked in large animals (Hackett 2011, shown above)   

4) Section 1.1 last paragraph: just adding transposase without having the cargo flanked by transposon inverted terminal repeats will not do any good. A good reference here is: Aronovich EL, McIvor RS, Hackett PB. The Sleeping Beauty transposon system: a non-viral vector for gene therapy. Hum Mol Genet. 2011 Apr 15;20(R1):R14-20. doi: 10.1093/hmg/ddr140.  For a detailed description of what happens with respect to gene expression and transposition in mouse liver,similar to the detail in this manuscript, following HT to mouse liver: Bell JB, Aronovich EL, Schreifels JM, Beadnell TC, Hackett PB. Duration of expression and activity of Sleeping Beauty transposase in mouse liver following hydrodynamic DNA delivery. Mol Ther. 2010 Oct;18(10):1796-802. doi: 10.1038/mt.2010.152.

5) Section 1.2: A good summary (a bit dated on liver gene therapy via Sleeping Beauty is: Izsvák Z, Hackett PB, Cooper LJ, Ivics Z. Translating Sleeping Beauty transposition into cellular therapies: victories and challenges. Bioessays. 2010 Sep;32(9):756-67. doi: 10.1002/bies.201000027. Erratum in: Bioessays. 2011 Jun;33(6):478-9.

6) Section 4.4: As alluded to above, HT is great for use in mice; but not in any larger animals (including small puppies), even with devices that can deliver nucleic acid preparations 

Reviewer 2 Report

Hydrodynamic injection in small animal models have been used to “screen” the potential of genes of interest, with the uptaking mechanism remained unclear. This manuscript reported ultrastructural changes by EM in mice model after Hydrodynamic delivery and expand the knowledge to the readers in the field.

Some revisions needed to be addressed.

 ·         Zhou et al. showed that the highest level of a reporter gene can be appreciated in the liver [30]: be specific, what kind of gene…

·         Reports show that up to 500 - 1,000 μg per ml of
serum and 45 μg of cellular protein/gram of liver are the highest product level reported by a single hydrodynamic injection into the mouse tail vein [31,33,35]: It is hard to follow what kind of points the authors wish to discuss

·         It has been used to express haemophilia factors [47]: Please check whether the citation is correct, Factor IX or Factor VIII?  

·         Materials and Methods: 2.1 Highly inbred 6-weeks-old male C57BL/6J/129Sv mice (n = 20; 25-30 g body weight) were purchased from Charles River Laboratories (Sulzfeld, Germany). There is a repeated statement, also at 6wk, this strain of mice are usually <25g.

·         Neck fracture: please use “cervical dislocation

Reviewer 3 Report

This is a very interesting study. Using electron microscope, the authors found that plasmid transfer by hydrodynamic transfection may be via endocytosis. In general, the preliminary findings of this study are valuable. The author should further check whether the text is edited correctly. For example, 2.1 Animal Studies seems to be repeated. I suggest the manuscript can be acceptable after minor revision.
